# Importance of EU Integration for Biodiversity and Nature Conservation in Transboundary Protected Areas (TPAs) in the Western Balkan

Aleko Miho [1,*], Jani Marka [1] and Zenel Krasniqi [2]

[1]   Department of Biology, Faculty of Natural Sciences, University of Tirana, Bulevardi Zogu I, 25/1, 1057 Tiranë, Albania
[2]   Faculty of Education, University "Hasan Prishtina", Rr. "Agim Ramadani", 10 000 Pristina, Kosovo
*    Correspondence: aleko.miho@fshn.edu.al

**Abstract:** There are many important protected areas in the Western Balkan region, which are shared between Albania, Montenegro, Kosovo, North Macedonia, and Greece. These areas have special importance based on their species density (mosses and higher plants) per surface unit. These transboundary ecosystems, which include mountainous massifs, lakes, and rivers, are biodiversity hotspots for the whole of Europe. Species and habitat densities are high compared to other countries in Southeast Europe. However, political borders fragment properly across two or three countries, which often have different approaches and rules for nature protection and the use of resources. Hence, in this short opinion piece, we stress common and cooperative transboundary protection and management in these countries. Furthermore, the European Union's policy towards the Western Balkan countries in the Stabilization and Association Process (SAP), with the goal of their eventual EU membership, is crucial. Therefore, our appeal is addressed not only to the respective local communities and national governments but also to the European Commission and related EU institutions. We stress the importance of these cross-border ecosystems in the integration process.

**Keywords:** Western Balkan SAP; transboundary PAs; biodiversity hotspots; protection and management; EU integration

## 1. Broken Nature in the Western Balkan Region

Transboundary protected areas (TPAs) in the Western Balkan (WB) region are shared between Albania, Montenegro, Kosovo, North Macedonia, and Greece. They are well-known hotspots of biodiversity [1,2]. They include the Albanian alps (also known as Bjeshket e Nemuna or Prokletije), divided between Albania, Kosovo, and Montenegro; Shkodra/Buna (Skadar/Bojana), divided between Albania and Montenegro; Sharri (Scardus), Korab-Koritnik and Shebenik-Jabllanice, divided between Kosovo, North Macedonia, and Albania; Ohrid and Prespa Lakes, divided between North Macedonia, Albania and Greece (Prespa); and the Vjosa (Aoos) river, divided between Albania and Greece. Hence, these TPAs are fragmented by political borders across two or three countries, which often have different approaches and rules for nature protection and the use of natural resources. There are also different protection criteria in each country, different histories, and, of course, even different local names. The most important TPAs are listed in Table 1 and shown in Figure 1.

According to the data presented in Table 1, Albania is the country richest in TPAs, which covers more than 13% of Albanian territory, followed then by Kosovo (10.68%), North Macedonia (8.51%), Montenegro (4.1%) and Greece (0.68%).

**Table 1.** The most important TPAs of Albania and the neighboring countries of Montenegro, Kosovo, North Macedonia, and Greeceare listed in descending order of year of declaration in the respective countries. IUCN management categories [3]: SR, Strict Reserve (I); NP, National Park (II, IUCN Category); MNR, Managed Nature Reserve (IV); PL, Protected Landscape (V); RS, Ramsar Site; TBR, Transboundary Biosphere Reserve. Data are taken mostly from Wikipedia and other sources cited herein.

| Local/English Name, Protection Status (IUCN Category) | Surface in Hectares | Year of Declaration |
|---|---|---|
| **Albania (13.14% of the territory)** | | |
| Alpet e Shqiperise/Albanian Alps NP (II) (comprising Valbona Valley NP, Thethi NP, and Gashi River Reserve) | 82,844.7 | 2022 |
| Korab–Koritnik MNR (IV) | 53,850.0 | 2011 & 2022 |
| Lugina e lumit Vjose/Vjosa/Aoos River Valley MNR (IV) | 7989.5 | 2022 |
| Liqeni i Shkodres/Skadar Lake MNR (IV) | 24,049.8 | 2022 |
| Prespa/Prespa watershed NP (II) & RS | 9424.4 & 15,119.0 | 2013 |
| Ohrid-Prespa TBR | 94,728.60 | 2014 |
| Shebenik-Jabllanice NP (II) | 33,927.7 | 2008 |
| Bredhi i Hotoves-Dangelli/Hotova Fir-Dangelli NP (II) | 34,361.1 | 2008 |
| Lumi Bune-Velipoje/Buna/Bojana River-Velipoja LP (V) | 23,027.0 | 2005 |
| Liqeni i Shkodres-Lumi Bunes/Shkodra/Skadar Lake-Buna/Bojana River RS | 49,562.0 | 2005 |
| Butrinti/Butrinti watershed NP (II) | 8591.2 | 2005 |
| Kanali i Cukes-Butrint-Kepi i Stillos/Cuka-Butrinti-Stillo Cape RS | 13,500.0 | 2002 |
| Pogradec/Pogradeci/Ohrid watershed LP (V) | 27,323.0 | 1999 |
| **Kosovo (10.68%)** | | |
| Bjeshket e Nemuna NP (II) | 63,028 | 2012 |
| Malet e Sharrit/Sharr Mountains NP (II) | 53,272 | 2012 |
| Oshlaku SR (I) | 20.0 | 1961 |
| Maja e Arnenit/Arneni Peak SR (I) | 30.0 | 1960 |
| **Montenegro (4.1%)** | | |
| Ulcinj Salina MNR (IV) & RS | 1477 | 2019 |
| Prokletije/Bjeshket e Nemuna NP (II) | 16,630 | 2009 |
| Skadarsko jezero/Lake Skadar/Shkodra Lake NP (II) & RS | 40,000 & 20,000 | 1983 & 1995 |
| **North Macedonia (8.51%)** | | |
| Šar/Scardus/Sharr Mountains NP (II) | 54,214 | 2021 |
| Ohrid-Prespa TBR | 94,728.60 | 2014 |
| Ohridsko Ezero/Liqeni i Ohrit/Lake Ohrid RS | 25,205 | 2013 |
| Dojransko Ezero RS | 2696 | 2007 |
| Prespansko Ezero/Liqeni i Prespës/Lake Prespa RS | 18,920 | 1995 |
| Galicica NP (II) | 22,700 | 1958 |
| Mavrovo NP (II) | 78,000 | 1949 |
| Pelister NP (II) | 17,150 | 1948 |

**Table 1.** *Cont.*

| Local/English Name, Protection Status (IUCN Category) | Surface in Hectares | Year of Declaration |
|---|---|---|
| **Greece (0.68%)** | | |
| Prespes/Mikris Kai Megalis Prespas/Prespa lakes NP (II) | 21,061/32,403 | 1974/2009 |
| Pindos NP (II) | 3515/48,387 | 1966/2005 |
| Vikos-Aoos NP (II) | 9595 | 1973 |
| Lake Mikri Prespa/Little Prespa Lake RS | 5078 | 1975 |

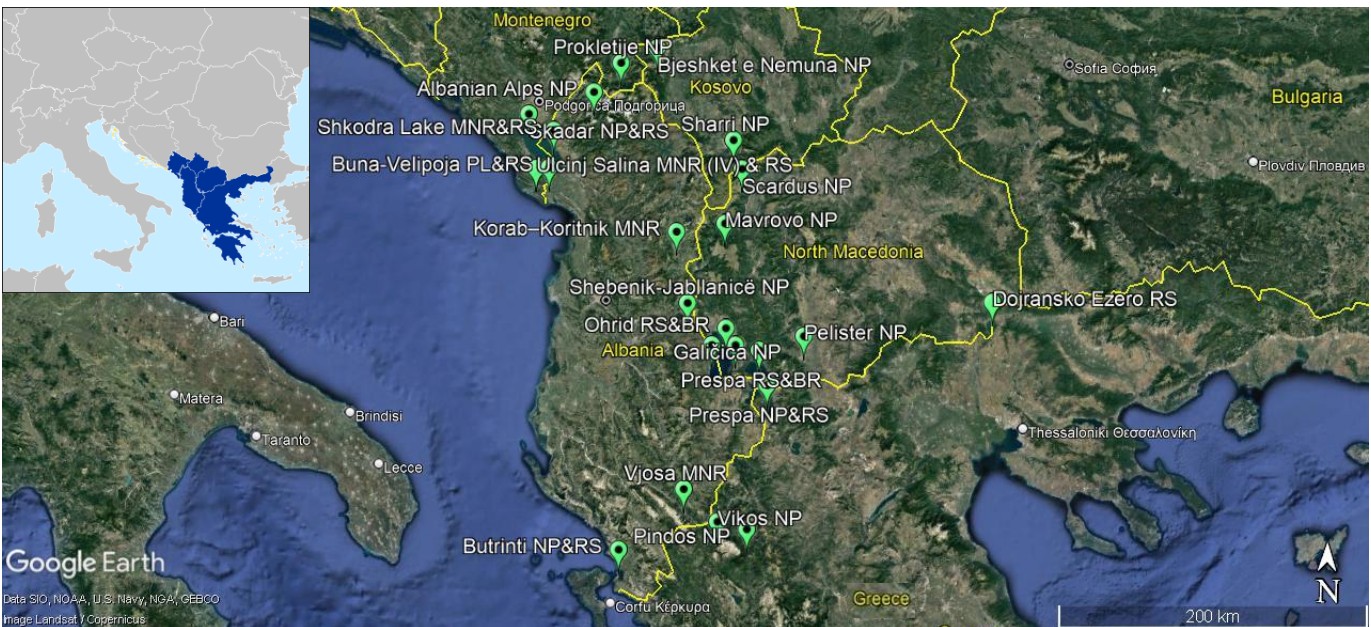

**Figure 1.** The most important TPAs of Albania and the neighboring countries of Montenegro, Kosovo, North Macedonia, and Greece (imagery date 14 December 2015). The names on the map are referred to the given names in the Table 1. IUCN management categories [3]: NP, National Park (II IUCN Category); MNR, Managed Nature Reserve (IV); PL, Protected Landscape (V); RS, Ramsar Site; BR, Biosphere Reserve. The maps are elaborated after Google Earth Pro, and the map in https://www.ncafp.org/the-balkans-at-a-crossroads/ (accessed on 14 December 2022).

**Shkodra/Skadar Lake** is partly protected as an NP (II) and RS in Montenegro, but it is protected as an MNR (IV) and RS in Albania. **Buna/Bojana River-Velipoja** is protected as an LP (V) and RS only in the Albanian part. Only **Ulcinj Salina** in the Montenegrin part of the Buna/Bojana delta is designated as an MNR and RS, which occurred recently (https://rsis.ramsar.org/ris/2399, accessed on 4 August 2019). The **Prespa Lakes** are protected as NPs (II) and RSs in Albania and Greece, and as an RS in North Macedonia (Figure 1).

The **Albanian Alps (Bjeshket e Nemuna/Prokletije)** constitute a cross-border massif of the Dinaric Alps between Kosovo, Albania, and Montenegro. In Kosovo, they have been protected as NPs since 2012, while the rest of the massif in Montenegro is also protected as an NP (2009) [4]. Meanwhile, the Albanian government recently designated the entire Albanian part of the Albanian Alps as an NP [5], including Bjeshket e Nemuna in Albania, the former Valbona Valley and Thethi NPs, and the Gashi River Reserve. We applaud this action and hope that it will help in further protecting the whole massif as a unique ecosystem, in close cooperation with Kosovo and Montenegro (Figure 1).

The **Sharr (Scardus/Šar) Mountains** are historically protected as NPs in Kosovo; the part of the mountains lying in North Macedonia was declared an NP in June 2021 [6]. In the south of the Sharr Mountains is the massif of **Mavrovo NP** in North Macedonia; while the massifs **Korab-Koritnik** extend into Albania, where they are protected as an MNR (IV). The **Shebenik-Jabllanice**, further south, is protected as an NP (Figure 1).

The whole **Vjosa River Valley** was recently designated an MNR (IV) by the Albanian government [7]. Furthermore, a working group led by the Ministry of Tourism and Environment has begun planning for the establishment of the Wild Vjosa River and its most important tributaries as an NP in Albania [8]; hopefully, it will be inaugurated in March 2023. Vjosa River is in ecological continuity with **Vjosa Delta-Narta Lagoon** LP (V) and **Hotova Fir-Dangelli NP** in Albania, and the **Vikos-Aoos** and **Pindos** NPs in Greece (Figure 1).

## 2. Biodiversity Hot Spots

### 2.1. Hot Spots in Mosses

The areas around the Mediterranean and the Black Sea harbor particularly high biodiversity, with many rare and endangered plant species and habitats [9,10]. The above-mentioned Western Balkan TPAs are particularly distinguished in this capacity. However, we do not make an overview of natural and biodiversity values here; rather, we discuss only some significant examples related to our research fields to show the importance of this region and the transboundary areas on conservation and sustainable management.

It is worth noting that the areas listed in Table 1 have attracted the interest of scientists at least since the second half of the last century. Their high biodiversity is due to their geographical position, origin, climate, etc.; together, they form variegated habitats where a rich living world is housed, including mosses that are important precursors of habitat formation.

In his PhD thesis, Krasniqi reported data on mosses (Bryophytes) from Kosovo, focusing mainly on two important protected areas: the Sharr and Bjeshket e Nemuna NPs [11].In this study, a total of 259 moss taxa were reported; 57 were newly reported mosses in Kosovo. There are ca. 360 total known moss species in Kosovo, with a density of 33 species/1000 km$^2$ (see Table 2). Of these, 168 species (or 47%) belong to the red list of mosses of Europe or the red lists of different Balkan countries [12].

**Table 2.** Numeric data on total moss species in some countries of SE Europe and related density per 1000 km$^2$. The names and related data from Albania and the neighboring countries, Montenegro, Kosovo, North Macedonia and Greece are bolded. Kosovo (Ko), Albania (Al), Bulgaria (Bg), Bosnia and Herzegovina (Ba), Greece (El), Croatia (Hr), North Macedonia (Mk), Montenegro (Me), Romania (Ro), Slovenia (Sl), Serbia (Rs), European Turkey (E-Tr), Southeast Europe (Se-Eu), Europe (Eu) [11].

| Countries | Ko | Al | Me | Mk | Rs | El | Ba | Bg | Hr | Ro | Sl | E-Tr | Se-Eu | Eu |
|---|---|---|---|---|---|---|---|---|---|---|---|---|---|---|
| **Species** | **359** | **459** | **555** | **345** | 556 | **525** | 462 | 719 | 476 | 749 | 640 | 217 | 899 | 1539 |
| **Species/1000 km$^2$** | **33** | **16** | **40** | **13** | 7 | **4** | 9 | 6 | 8 | 3 | 32 | 9 | 2 | 0.2 |

The moss species density in four WB countries (Ko, Al, Me, and Mk in Table 2) is higher than in other countries in Southeast Europe. Meanwhile, there are only 1539 known moss species in Europe, with a modest density of only 0.2 species/1000 km$^2$. The richness and diversity of mosses are mainly due to the presence of transboundary protected areas. For instance, for Kosovo, moss diversity is mainly due to the two mentioned mountainous massifs: the Sharri NP with a total of 255 known species, and the Bjeshket e Nemuna NP with a total of 239 species. Therefore, species richness and the harboring of rare and endangered moss species are quite evident in these TPAs.

*2.2. Hot Spots in Other Vegetal Groups*

There are about 3000 species of vascular plants in **Kosovo**, with up to 275 species /1000 km$^2$, while ca. 10% are threatened [13,14]. There are known already to be ca. 3650 vascular plant species in **Albania [15]**, with 130 species/1000 km$^2$. There are ca. 3350 vascular plant species in **North Macedonia** (https://en.wikipedia.org/wiki/ Flora_of_North_Macedonia, accessed on 6 June 2020), with ca. 130 species/km$^2$. Meanwhile, for the whole European continent, Silva et al. reported up to 12,500 vascular plant species, with only 1.2 species/1000 km$^2$ [9].

Here, the contribution of the mentioned transboundary NPs is quite relevant. For example, around 1000 vascular plants have been reported in **Bjeshket e Nemuna NP** in Kosovo, with 126 rare and endangered species [16]; ca. 1558 plant species have been reported in **Sharri NP**, with 107 species on the red list of Kosovo [17]. Furthermore, Hashani reported 997 plant species for the transboundary areas of Koritnik, Vraça, and Oshlak, which are in Kosovo as well [18]. As for North Macedonia, in his PhD thesis on the flora and vegetation of **Sharri Mountain** (North Macedonian part), Abdi reported about 2000 plant taxa, with ca. 460 rare or endangered species [19]. The flora of the **Albanian Alps** is considerably rich; to date, more than 1630 plant species have been reported (about 45% of Albanian flora), along with more than 20 habitats of international conservation interest [20].

Experts of the Hellenic Botanical Society (HBS)report that the vascular flora of Greece comprises 5927 species [21], or around 50 species/1000 km$^2$.Around 1800 vascular plant taxa have been recorded respectively in the Prespa National Park [22], and in the Vikos-Aoos National Park (NW Greece), many of them rare and unique to the area (https:// carpediemeire.com/2020/05/17/vikos-aoos-national-park/, accessed on 14 December 2022).

In the **Ohrid** and **Prespa lakes** and their watersheds, Levkov and Williams reported 919 diatoms: 789 taxa from Ohrid, with 117 endemics, and 244 taxa from Prespa, with 33 endemics [23]. For **Lake Shkodra/Skadar**, 930 species of algae and 497 vascular plant species have been reported; additionally, there are up to 1900 taxa of vascular plants in the whole Shkodra watershed [1].

**Vjosa/Aoos** has recently been considered a riverine ecosystem of European significance due to its biodiversity, hydromorphology, and sediment transport, as reported in several recent studies [24,25]. About 1430 plant and animal species are known to exist in the Albanian part of the Vjosa River; 39 species are endangered according to the IUCN, 148 are listed in Annex 1–3 of the Berne Convention, 41 are in the Bird Directive and 78 are in the Habitats Directive [26].

## 3. Ethnic Richness vs. Generous Biodiversity in Nature

These Western Balkan countries are also rich in history, ethnic groups, languages, religions, folklore, etc. The area has been a crossroads of many cultures and religions, where Bulgars and Slavs, Orthodox Christians, Catholics, and Muslims have long met [27]. Over centuries, different empires have left their traces, such as the Roman, Byzantine, Bulgarian, Ottoman, and Austro-Hungarian. Today's states were formed in the last centuries, with more or less sharp political borders. We, humans, must be inspired by the generosity of nature and the coexistence of species and habitats that these natural areas offer. Along with the data discussed above on the species density of mosses and vascular plants, let us bring another example from the microscopic world; in a periphyton sample or a few drops of the related cleaned material from the Ohrid Lakeshore in Tushemishti (Pogradeci) more than 150 species of diatoms were found, including species new to science [28,29].

Natural richness and generosity have helped the survival of local ethnicities, providing food, clothing, health, and shelter, but have also provided protection from invasions over the centuries. The Shkodra/Skadar and Ohrid towns are the most ancient settlements of the Western Balkans. All the areas mentioned in Table 1 are commonly visited nowadays; they offer opportunities in all aspects of life and leisure, such as bathing, boating, fishing, hiking and climbing, skiing, hunting, traditional cuisine, etc. Many visitors enjoy a trip to

Shkodra/Skadar, Ohrid, and Prespa areas, in the Valbona, Shala, Cemi or Vjosa/Aoos river valleys, in the Šar/Scardus/Sharr, Korabi, Galičica or Pelister mountains, etc. The local flora is also important to household medicine and food security [19,20,30,31]. These are but a few examples.

However, this natural wealth does not seem to align with the well-being of nations; as a whole, the region remains the poorest in Europe. Internal problems, either inherited from history or recently formed, hold the economy and development hostage, and often even affect fruitful cross-border cooperation. In many cases, these problems have caused negative pressure and impacts on the preservation of natural and biological values, including in TPAs. For instance, it is worth mentioning the construction of tourist infrastructure even in the most sensitive areas, deforestation, poor land use, and fires [1,30]. Furthermore, negative impacts from agriculture, fishing, and aquaculture, and overexploitation or unsustainable harvesting of medicinal and aromatic plants have also been reported for Albania and North Macedonia [31–33]. Finally, the construction of hydropower plants (HPPs) over the last decade has had a large impact on hundreds of rivers, including the Valbona river NP [30,34,35]. These problems may have negative consequences for each country, quality of life, and the climate, and may require costly restorative measures for future generations. The recent EU 2022 Enlargement Package reported moderate or limited progress in the areas of environment and climate change, especially for Albania and Kosovo [36].

## 4. Transboundary Protection of Unique Ecosystems Is Needed

Richness in habitats and plant and animal species is an important resource, from both natural and economic perspectives. However, it also implies the responsibility to continuously preserve nature, habitats, and wildlife. Careful use, preservation, and continuous restoration are the responsibilities of each neighboring country, alongside the need to work in close cooperation with each other and with international experts and institutions. This also means that ecosystem services must be always properly considered, as recognized by EU policy and international environmental conventions on the environment.

Hence, we emphasize the importance of common transboundary protection and management of unique ecosystems. Permanent common efforts would help to better preserve these valuable ecosystems. TPAs can serve as bridges of cooperation for research and allow countries to join forces in protection and restoration measures. EU integration is crucial in joint plans for sustainable cross-border development in tourism, forestry, agriculture and livestock, medicinal plants, handicrafts, hunting, tourism, etc.

These joint efforts started after the 1990s in WB countries, e.g., for Prespa Lakes and the related Prespa NPs [37–39], divided between Albania, Greece, and North Macedonia; for the Ohrid, divided between Albania and North Macedonia [40]; and for the Shkodra Lake basin, divided between Albania and Montenegro [41]. Recently, efforts have been made to promote transboundary cooperation for the Vjosa/Aoos wild river NP in Greece and Albania [42]. Nevertheless, these cooperations have been relatively sporadic and often short in duration, along with facing various other challenges.

In our opinion, the respective WB countries should proactively engage with each other to tackle eventual problems and embrace a better environmental approach for sustainable development and growth. We stress the **importance of socio-ecological studies** and close cooperation between the academic world, decision-makers, and investors in environmental sustainability. The quality of knowledge of all actors and the **Science–Policy interface** is crucial in properly solving environmental problems and balancing development with the conservation of natural resources [25,43].

In this transboundary context, we agree that ethnic richness can play an important role through the traditional use of natural resources [31,33] and can support transboundary nature conservation. Moreover, this richness is also crucial to rise above ethnic controversies whenever they exist. Many examples in Europe and the world show that political borders, whenever they are established or strengthened, not only obstruct the mutual development and growth of each country but also lead to the misuse or harm of shared natural resources.

In this context, the European Union's policy and the Stabilization and Association Process (SAP) are crucial [36]. Our appeal is addressed not only to the respective national governments in the SAP process but also to the European Commission and related EU institutions, which should consider the importance of the unified protection of these cross-border ecosystems in the integration process. Gardin confirms '*the experience of the international Prespa Park may serve as an experimental essay in realizing the integration of marginalized borders of the EU through the environment*' [38]. Joint protection and management of TPAs with high natural value in Western Balkan areas may also lead to a new approach to the self-determination of EU borders in the future.

**Author Contributions:** Conceptualization, A.M.; methodology, A.M. and J.M.; validation, A.M., J.M. and Z.K.; formal analysis, A.M.; investigation, J.M. and Z.K.; resources, J.M. and Z.K.; data curation, A.M., J.M. and Z.K.; writing—original draft preparation, A.M.; writing—review and editing, A.M.; visualization, A.M. and J.M.; supervision, A.M. and J.M.; project administration, none; funding acquisition, none. All authors have read and agreed to the published version of the manuscript.

**Funding:** This research received no external funding.

**Institutional Review Board Statement:** Not applicable.

**Informed Consent Statement:** Not applicable.

**Data Availability Statement:** The manuscript is ethically sound and meets industry-recognized standards that are reflected in MDPI policies.

**Conflicts of Interest:** The authors declare no conflict of interest.

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
