# Peer review of "Importance of EU Integration for Biodiversity and Nature Conservation in Transboundary Protected Areas (TPAs) in the Western Balkan"

_2673-9917, doi:10.3390/hydrobiology2010015_

Round 1
Reviewer 1 Report
The authors used existing published data and made a strong case for protecting this biodivers environment and strengthening the existing TPAs.
Author Response
Nothing to comment! We did our best to revise and improve the original manuscript.

Reviewer 2 Report
In general, this paper conveys an interesting idea to preserve conservation areas located across countries in a unified landscape with high conservation value. Using the scope of the Hydrobiology journal as a guide, the approach to integrating protected area management should focus on wetland ecosystems and use terrestrial ecosystems as support.
From an ecological point of view, it's also important to explain why integrated management of cross-border conservation areas is essential, whether it's as a migration route, a stepping stone for migrating wildlife species, or for other important reasons.
Line 1: The title and substance of this paper are not within the scope of the journal Hydrobiology.
Line 2: It is necessary to explain what "EU" stands for.
Line 9-10: Examples of conservation areas need not be mentioned in the abstract.
Line 19: The abbreviation PAs is mentioned first.
Line 20: The abbreviation "European Union" is mentioned first.
Line 30: Reference to the area as a biodiversity hotspot is needed. Read Myers et al. (2000). Biodiversity hotspots for conservation priorities. Nature 403
Lines 40-41: Cite the IUCN Category source.
Line 62: Non-standard map, no coordinates, and a global map insert. The area of protected area must be shown on the map, not only at location points. That is important to determine what connectivity is needed, such as wildlife migration routes or stepping stones for birds. The PAs map area can be found at: https://www.protectedplanet.net/en/thematic-areas/wdpa?tab=WDPA
Lines 47-60 and 67-77: This description will be easy to understand if you refer to the PAs area on the map.
Line 62: Map needs a global insert
Line 78: Biodiversity is not only at the species level but also at the ecosystem and genetic level. Each should be explained here.
Line 79-108: Discussing species of mosses at length is not relevant to the biodiversity context. because it is not an indicator of biological species diversity like birds, for example.
Line 108-132: The data in this narrative looks complicated; change it to tabulated form and only narrate the most important ones.
Lines 11, 33, 86, 129, 135, 144, 158, 159, 169: Avoid using "etc." it is very vague and unscientific.
Line 202: Writing a bibliography needs to be more careful and according to the guidelines. doi links are included. Prioritize primary reference sources, secondary and unpublished references are restricted for use.
Line 242: Add doi in all cited journals.
Author Response
Line 1: The title and substance of this paper are not within the scope of the journal Hydrobiology.
Author’s answer: This opinion was originally a presentation (as invited speaker) in Ablakes 4, in Elbasani University, Albania. We thought it would help if edited at the Special Issue "Building Resilience of Water Ecosystems through Scientific Knowledge”. Ours is a modest opinion or an open letter, not purely scientific, addressed to local communities and further European Community institutions for the importance that the whole region offers. Let us mention Ohrid, Prespa and Shkodra Lakes, or Buna, Vjosa and other distinguished Rivers, unique ecosystems chopped by the political borders.
It is not a review about biodiversity; it was not our interest, and also beyond to our abilities. During preparation of the PhD on Kosovo mosses we were excited by the fact of high species density for 1000 km2 [7]. Some years ago Fremuth (ed.) in the Albania's Guide to its Natural Treasures [42], as coauthors (Miho) we have confirmed that Albania is an exceptional country within Europe regarding flora and vegetation with 30% of Europe’s flora in an area of only 0.27% of Europe. It was often mentioned as a resource and a responsibility towards conservation at the same time. Here we added few more data to this concept of species density, even from the neighboring countries of Albania, wishing to motivate and stir up further the integration process, and the management of these ecosystems as whole. We wish that our countries to be integrated soon in EU conserving the high biodiversity values. But if the present plans of building more than 500 HPPs in Albanian rivers [43] there will be about 20 HPPs/1000 km2, and much less species of plants and animals for this same unit area. This example towards the present approach in nature conservation in our countries, and other mentioned here and elsewhere, are an urgent challenge. It would be never late for the change!
- Fremuth, W. (Ed.), Albania Guide to its Natural Treasures, 2000. Herwig Klemp/ECAT Tirana. 140 pp. ISBN: 3-931323-06-4
- Çibuku, A., “Të dëmshme dhe të padobishme”, rreth 1,100 hidrocentrale mbi lumenjtë e Shqipërisë dhe Bosnje Hercegovinës. January 31, 2023. https://citizens-channel.com/2023/01/31/te-demshme-dhe-te-padobishme-rreth-1100-hidrocentrale-mbi-lumenjte-e-shqiperise-dhe-bosnje-hercegovines/
Line 2: It is necessary to explain what "EU" stands for. Ok!
Line 9-10: Examples of conservation areas need not be mentioned in the abstract. Ok!
Line 19: The abbreviation PAs is mentioned first. Ok!
Line 20: The abbreviation "European Union" is mentioned first. Ok!
Line 30: Reference to the area as a biodiversity hotspot is needed. Read Myers et al. (2000). Biodiversity hotspots for conservation priorities. Nature 403 Ok!
Lines 40-41: Cite the IUCN Category source. Ok!
Burhenne-Guilmin, F., Guidelines for Protected Areas Legislation. IUCN, 2011. p. 147. ISBN 9782831712451.
Line 62: Non-standard map, no coordinates, and a global map insert. The area of protected area must be shown on the map, not only at location points. That is important to determine what connectivity is needed, such as wildlife migration routes or stepping stones for birds. The PAs map area can be found at: https://www.protectedplanet.net/en/thematic-areas/wdpa?tab=WDPA
We consider not proper necessary, and then the PAs in www.protectedplanet.net are not so updated.
Lines 47-60 and 67-77: This description will be easy to understand if you refer to the PAs area on the map. Ok!
Line 62: Map needs a global insert OK!
Line 78: Biodiversity is not only at the species level but also at the ecosystem and genetic level. Each should be explained here. It is not a review about biodiversity; it was not our interest, and also beyond our abilities. See explanations given above.
Line 79-108: Discussing species of mosses at length is not relevant to the biodiversity context. because it is not an indicator of biological species diversity like birds, for example. Explanations given above.
Line 108-132: The data in this narrative looks complicated; change it to tabulated form and only narrate the most important ones. Not easy to gather in a table!
Lines 11, 33, 86, 129, 135, 144, 158, 159, 169: Avoid using "etc." it is very vague and unscientific. We changed something, but let us stress the aspect of not purely scientific aspect of our opinion.
Line 202: Writing a bibliography needs to be more careful and according to the guidelines. doi links are included. OK! Prioritize primary reference sources, secondary and unpublished references are restricted for use.
Line 242: Add doi in all cited journals
For some local journals we do not find a DOI!

Reviewer 3 Report
I appreciate the concept that the authors have introduced though the paper “Importance of EU integration for the biodiversity and nature 2 conservation of trans-boundary Western Balkan Pas”. However few comments are pointed which may improve your manuscript.
1.It would be better if you concise the abstract into a single paragraph.
2. Some short forms such as PA and SE need to be expanded at least once in the entire text, perhaps the first time it is mentioned. Similarly, if you want to use the short form in the rest of the text, insert the short form in brackets along with its extension, the first time it is mentioned. Furthermore, if you use the shortform once in the text, please continue with it throughout the text ( eg, WB).
3. In Table 1, the given data seems random. It would be better if you arrange them either in alphabetical order of their name, in descending order of the surface area, or in descending order of year of declaration, under each territory.
4. In the lines between L44- L46 data regarding Greece is missing.
5. The text references for L100 - L106 is missing.
6. There is a typological error in Line-72. Please correct it as ‘designated’.
7. Please rephrase the sentence (Line-79-81, L-153-157) or avoid mentioning the author’s name within the text since reference numbers are already given.
8. Rephrase sentences (L-89, L-112, L-117, L-123), check for the grammatical errors, mentioning the author’s name within the text since reference numbers are already given.
9. Please check for the grammatical errors in L-125.
10. The authors have written ‘’ European significance [19, 20, etc.]’’ in L-129, ‘’ 2014-2022 [4, 20]; etc. ‘’ in L-158 and “Kosovo, too [7, 10; etc.]” in L-159. What does it mean adding etc. to the reference list?
11. Please avoid ‘ie.’ from the reference list (L-136, L-172).
12. The whole manuscript need revision for typological and grammatical errors.
13. Although you call for EU integration for biodiversity and nature, you do not mention the disadvantages of current conservation strategies and your possible proposals. It would be better if you add your own views to the paper in addition to the data from previous studies.
14. The year of publication is missing in ref.no 8, in the given format.
Author Response
1.It would be better if you concise the abstract into a single paragraph. Done!
- Some short forms such as PA and SE need to be expanded at least once in the entire text, perhaps the first time it is mentioned. Similarly, if you want to use the short form in the rest of the text, insert the short form in brackets along with its extension, the first time it is mentioned. Furthermore, if you use the short form once in the text, please continue with it throughout the text ( eg, WB). Done!
- In Table 1, the given data seems random. It would be better if you arrange them either in alphabetical order of their name, in descending order of the surface area, or in descending order of year of declaration, under each territory. Listed in descending order of year of declaration, under each territory!
- In the lines between L44- L46 data regarding Greece is missing. Done!
- The text references for L100 - L106 is missing. Added!
- There is a typological error in Line-72. Please correct it as ‘designated’. Corrected!
- Please rephrase the sentence (Line-79-81, L-153-157) or avoid mentioning the author’s name within the text since reference numbers are already given. Hope is ok!
- Rephrase sentences (L-89, L-112, L-117, L-123), check for the grammatical errors, mentioning the author’s name within the text since reference numbers are already given. Hope is ok!
- Please check for the grammatical errors in L-125. Hope is ok!
- The authors have written ‘’ European significance [19, 20, etc.]’’ in L-129, ‘’ 2014-2022 [4, 20]; etc. ‘’ in L-158 and “Kosovo, too [7, 10; etc.]” in L-159. What does it mean adding etc. to the reference list? Somehow explained!
- Please avoid ‘ie.’ from the reference list (L-136, L-172). Corrected!
- The whole manuscript need revision for typological and grammatical errors. Corrected!
- Although you call for EU integration for biodiversity and nature, you do not mention the disadvantages of current conservation strategies and your possible proposals. It would be better if you add your own views to the paper in addition to the data from previous studies. We added something: In our opinion, our countries should do efforts to solve as soon as possible the eventual problems and embrace a better environmental approach on the way of development and growth. Here we would stress the importance of socio-ecological studies and close cooperation between the academic world, decision-making and investors in environmental sustainability. The quality of knowledge of all actors in this process and the Science-Policy interface are crucial in balancing better the development with the conservation of natural resources [20, 34]. In transboundary context, it would be quite helpful finding and joining the common forces to rise above ethnic contrasts historically inherited, when they exist. Many examples today in Europe and the World show that political borders, whenever they are established or strengthened, not only obstacle mutual development and growth of each country, but also misuse or damage common natural resources.
- The year of publication is missing in ref.no 8, in the given format. Corrected!

Reviewer 4 Report
Some suggestions are enclosed in the attached paper comments.

Author Response
- How people in surrounding interact with the PAs is not clearly described in this article. I assume if there is no any disturbance by people activities, and the areas are still intact. Thus, at least authors explain the current activities across the areas that threats the ecosystems and all flora and fauna exist in the areas. Added something more in Chapter 3: However, all this natural wealth is contrary to the well-being of each nation; the region continues to remain as a whole among the poorest in Europe; internal problems, inherited from past history, or newborn, keep hostage economy and development, and often even the fruitful cross-border cooperation. Not in few cases, it has caused negative pressure and impact towards the preservation of natural and biological values, including on TPAs; worth mentioning the building the tourist infrastructures even in the most sensitive areas, deforestation for urban and tourist buildings; unfriendly impact from agriculture, fishing and aquaculture; not sustainable harvesting of aromatic medicinal and aromatic plants; and last, but not least, the large impact of the last decade for the construction of HPPs in rivers, even on Valbona river NP [26, 27]. Last EU 2022 Enlargement Package report moderate or limited progress in the areas of energy, environment and climate change, especially for Albania and Kosovo [35].
- In related to human activities, from this perspective readers also expect to obtain information on how the natural and biological resources contribute to the local livelihood, as human also rely on. Added something more in Chapter 3: Of course, the nature richness and generosity has helped the survival of the local ethnities, providing with food, clothing, shelter, but also protection from the invasions in centuries. Shkodra/Skadar and Ohrid towns belong to the most ancient settlements of the Western Balkans. All the mentioned areas in Table 1 are the most visited nowadays; they offer pleasure in all life aspects, bathing, boating, fishing, hiking and claiming, skiing, hunting, traditional cuisine, etc. Many of us enjoy a trip in Shkodra/Skadar, Ohrid and Prespa lakes, in Valbona, Shala, Cemi, Vjosa/Aoos river valleys, in Šar/Scardus/Sharr, Korabi, Galičica or Pelister Mountains, etc. Just to bring few examples.
- Rich ethnicity vs. generous biodiversity of Mother Nature (line number 133), author should explain how the diversity of ethnic groups including religions play important roles to rule the use of natural resources and the traditional ecological knowledge of the people support the nature conservation as part of human interactions with nature. This is important, so to further speed up the nature conservation of the transboundary, this will be integrated with science approaches point 4 of the articles. Many thanks for the suggestion! We added something in Chapter 4: In transboundary context, we completely agree with progressive thought that rich ethnicity can play an important role in the traditional use of natural resources [about plants in 23, 35]; the traditional ecological knowledge of different ethnic groups can serve as a support for nature conservation, and could further accelerate transboundary nature conservation.
- Any efforts from each nation across the border (authors should mention them), and later can be elaborated with what authors suggested. We mention something in Chapter 4: Such joint efforts started after 1990s for Prespa Lakes and the related Prespa NPs [28], between Albania, Greece and North Macedonia; for the Ohrid between Albania and North Macedonia [29]; for the Shkodra Lake basin between Albania and Montenegro [30]; international and local environmental institutions are making efforts to promote the transboundary cooperation for Vjosa/Aoos wild river NP in Greece and Albania [33]. Nevertheless, these efforts are relatively sporadic, not always durative, and separated by more or less political borders and views.
- Elaborated all the stuffs, put in the scenario resulted from this opinion will enrich the manuscript for readers. We did our best!

Reviewer 5 Report
The opinion of Miho and his colleagues on the importance of EU integration for the biodiversity and nature conservation of transboundary Western Balkan PAs though sounds optimistic the article however lacks clarity. The article is verbose in nature with lots of sentences without any clear meaning to the reader. In my opinion, the article has just simply highlights the list of PAs situated in Balkan states along with list of species (primarily plants) present in each of these PAs.
I would be happy had they included other organism especially higher vertebrates as this normally attracts conservationists’ attention and convince one to agree to the idea of conservation unitedly.
The authors could have highlighted the economic value(s) of plant species present in each PAs as this helps different states to share their revenue on the mutually agreed terms and conditions. In the present day scenario just conserving species (especially medicinal plants) without locals benefiting out of such conservation efforts may not be successful in long run. Successful utilization of existing resources between range states on mutually agreed terms and conditions will make this initiative meaningful and local will support such initiatives as the authors have mentioned that different ethnic groups, languages, religions represent each country in the region. How to make them agree on a single or a set of points should have been the crux of the article and hoe they address such issue should have been main opinion of the authors.
The moment we talk about sustainable utilization and harvesting of medicinal plants and other plants of economic importance between Balkan nations, the CITES comes into play. The authors should have addressed this aspect too while drafting the article.
Author Response
Many thanks for your comments and suggestions! We did our best to revise the manuscript and improve it somehow.

Round 2
Reviewer 2 Report
ok, all suggestions clarified.
Author Response
Dear Colleague, we did my best to answer as suggested:
-Explained some abbreviations, starting from the title and also during the text; we do not think it would be necessary to have a table for the abbreviations; for most there are given explanations in respective titles of tables and figure.
-Changed the letter 'ë' in 'e' (we hope this was suggested!);
-Added the date of the map access in the figure 1.
We accepted all the track changes and uploaded the final version. We are grateful to the reviewers for their valuable comments and suggestions! And of course many thanks to your patient and friendly way in assisting us!
With my best regards
Aleko Miho

Reviewer 5 Report
Thanks for addressing the issues that I have raised and now the article is clear in terms of language correction and improvement.
Author Response
Dear Colleague, we did my best to answer as suggested:
-Explained some abbreviations, starting from the title and also during the text; we do not think it would be necessary to have a table for the abbreviations; for most there are given explanations in respective titles of tables and figure.
-Changed the letter 'ë' in 'e' (we hope this was suggested!);
-Added the date of the map access in the figure 1.
We accepted all the track changes and uploaded the final version. We are grateful to the reviewers for their valuable comments and suggestions!
With my best regards
Aleko Miho
